# Metabolomic Profiling of Mice with Tacrolimus-Induced Nephrotoxicity: Carnitine Deficiency in Renal Tissue

**DOI:** 10.3390/biomedicines12030521

**Published:** 2024-02-26

**Authors:** Sho Nishida, Tamaki Ishima, Natsuka Kimura, Daiki Iwami, Ryozo Nagai, Yasushi Imai, Kenichi Aizawa

**Affiliations:** 1Division of Clinical Pharmacology, Department of Pharmacology, Jichi Medical University, Tochigi 329-0498, Japan; snishida@jichi.ac.jp (S.N.); ishima.tamaki@jichi.ac.jp (T.I.); kimura_n@jichi.ac.jp (N.K.); imaiy@jichi.ac.jp (Y.I.); 2Division of Renal Surgery and Transplantation, Department of Urology, Jichi Medical University, Tochigi 329-0498, Japan; iwamid@jichi.ac.jp; 3Jichi Medical University, Tochigi 329-0498, Japan; rnagai@jichi.ac.jp; 4Clinical Pharmacology Center, Jichi Medical University Hospital, Tochigi 329-0498, Japan; 5Division of Translational Research, Clinical Research Center, Jichi Medical University Hospital, Tochigi 329-0498, Japan

**Keywords:** metabolome, tacrolimus, chronic kidney disease, metabolite, histamine, tacrolimus induced nephrotoxicity, antioxidant, carnitine, transplantation

## Abstract

Tacrolimus (TAC)-induced chronic nephrotoxicity (TAC nephrotoxicity) has a detrimental effect on long-term kidney graft survival. However, the pathogenesis of TAC nephrotoxicity remains largely unknown. We explored it by focusing on metabolic changes in renal tissues. In this study, mice were separated into TAC and control groups (n = 5/group). TAC was administered to the TAC group (1 mg/kg/d for 28 days) subcutaneously. The control group was similarly treated with normal saline. Renal tissue metabolomes were evaluated. Renal fibrosis was observed only in the TAC group. Metabolomic analysis showed that carnitine and related metabolites were substantially lower in the TAC group than in the control group, presumably due to impaired biosynthesis and reabsorption. Low carnitine levels impair antioxidation in renal tissues and β-oxidation in mitochondria, which may lead to renal tissue damage. This metabolomic analysis revealed that carnitine deficiency in renal tissue appears to explain TAC nephrotoxicity.

## 1. Introduction

Tacrolimus (TAC), a calcineurin inhibitor (CNI), has been an extremely effective immunosuppressive agent for successful organ transplantation and autoimmune disease suppression. In the field of kidney transplantation, TAC largely suppresses acute rejection and has markedly improved short-term graft survival [1]. Thus, approximately 90% of maintenance immunosuppressive management in the field of kidney transplantation today is performed using TAC and mycophenolate mofetil [2]. On the other hand, various types of organ damage and adverse reactions result from long-term oral administration of TAC [3]. Among these adverse reactions, patients taking TAC almost inevitably develop chronic kidney injury within 10 years [4]. Indeed, many solid organ transplant patients subsequently progress to end-stage renal disease [5,6]. Therefore, this chronic kidney disease must be resolved to achieve long-term kidney graft survival.

TAC-induced chronic nephrotoxicity (TAC nephrotoxicity) is defined as irreversible and progressive loss of transplant renal function due to long-term exposure to TAC [3]. Pathogenesis of TAC nephrotoxicity has been attributed to production of reactive oxygen species (ROS) in proximal tubules, resulting in impaired antioxidation [7], and a mixture of chronic hemodynamic compromise and direct tubular injury due to damage to micro-vessel walls [3]. Diagnosis of TAC nephrotoxicity is made by transplant kidney biopsy, with characteristic pathologic findings of CNI-specific hyalinization of arterioles, and interstitial and tubular atrophy with fibrosis [8]. However, detailed pathogenetic mechanisms are largely unknown. Several treatments, such as calcium channel blockers and angiotensin receptor antagonists have been used to ameliorate renal dysfunction caused by TAC nephrotoxicity, but to date, no effective results have been obtained to alleviate progression of this disease [3]. Therefore, TAC is used continuously in clinical practice with tacit acceptance of its chronic adverse effects.

Recently, pathological analysis focused on metabolites offers new hope for discovering pathophysiological mechanisms of chronic kidney disease and primary illness. Metabolomic analysis of human samples and animal experimental models in each disease is an innovative way to find biomarkers and targets for therapeutic intervention [9,10,11]. This approach will reveal the pathophysiology of TAC nephrotoxicity, but to date, there has been no report of comprehensive metabolomic analysis. In this study, we attempted to better understand pathogenesis of TAC nephrotoxicity by focusing on metabolic changes in renal tissues, using mice with chronic nephrotoxicity induced by TAC administration.

## 2. Materials and Methods

### 2.1. Animals and TAC Administration Protocol

Seven-week-old male ICR mice were fed a low sodium diet (0.01% sodium, CLEA Japan, Inc., Tokyo, Japan) ad libitum. After seven days, mice were divided into two groups. Osmotic pumps (ALZET^®^ osmotic pump 2004, ALZET Osmotic pumps, Cupertino, CA, USA) in the TAC group were filled with TAC (Prograf^®^, Astellas Pharma, Inc., Tokyo, Japan) at 1 mg/kg/day, calculated from the flow rate, whereas osmotic pumps for the control group were filled with normal saline. After 40 h of incubation in saline at 37 °C, pumps were implanted subcutaneously in the backs of mice for continuous administration. After 28 days of continuous subcutaneous administration, mice were weighed before renal tissue collection. Renal tissue collection was performed under isoflurane anesthesia (induction concentration 4–5%, maintenance concentration 2–3%). This protocol was performed as previously reported [12,13] (Figure 1).

### 2.2. Pathological Analysis

Kidney samples were fixed with 4% paraformaldehyde (Nacalai Tesque, Inc., Kyoto, Japan) and then embedded in paraffin. Tissues were cut into 5-µm slices and then treated with Masson trichrome stain.

### 2.3. Quantitative PCR Analysis

QPCR analysis was performed using the following steps. Briefly, first, total RNA was extracted from cryopreserved kidney tissue using an RNeasy^®^ Mini kit (QI-AGEN, Venlo, The Netherland) according to the product protocol. Second, total RNA was converted to cDNA using Revertra Ace^®^ qPCR RT Master Mix with gDNA Remover (Toyobo Co., Ltd., Osaka, Japan). Finally, PCR was performed using THUNDERBIRD^®^ Next SYBR^®^ qPCR Mix (Toyobo Co., Ltd., Osaka, Japan) and PCR conditions comprised an initial heating step at 95 °C for 3 min followed by 35–40 cycles of a two-step reaction (95 °C for 3 s, 60 °C for 20 s). Glyceraldehyde-3-phosphate dehydrogenase (*GaPDH*) was used as the endogenous standard gene. Comparative CT (ΔΔCT) values and relative values with respect to the control group were calculated from the obtained CT values using Excel. Primer sequences used for this qPCR were as follows.

*Acta2*: Forward; AGTGTGATATTGACATCAGGAAGGA, Reverse; ACAGAGTACTT-GCGTTCTGGGAG, *Kim-1*: Forward; TCCACACATGTACCAACATCAA, Reverse; GTCACAGTGCCATTCCAGTC, *Tgfb1*: Forward; ACTGGAGTTGTACGGGCAGTG, Reverse; GGCTGATCCCGTTGATTTCC, *GaPDH*: Forward; TGTGTCCGTCGTG-GATCTGA, Reverse; TTGCTGTTGAAGTCGCAGGAG.

### 2.4. Metabolome Analysis

Approximately 20–40 mg of frozen tissue were placed in a homogenization tube, along with zirconia beads (5 mmφ and 3 mmφ). Next, 600–750 µL of 50% acetonitrile/Milli-Q water containing internal standards (Methionine Sulfone and D-camphor-10-sulfonic acid) [14] were added to the tube, after which the tissue was completely homogenized at 1500 rpm, 4 °C for 120 s using a beads shaker (Shake Master NEO, Bio Medical Science, Tokyo, Japan). The homogenate was then centrifuged at 2300× *g*, 4 °C for 5 min. Subsequently, the upper aqueous layer was filtered through a 5-kDa-cutoff Millipore centrifugation filter (UltrafreeMC-PLHCC, HMT, Tsuruoka, Japan) at 9100× *g*, 4 °C for 120 min to remove macromolecules. The filtrate was evaporated to dryness under vacuum and reconstituted in Milli-Q water for metabolomic analysis.

Metabolome analysis was conducted using capillary electrophoresis Fourier transform mass spectrometry (CE-FTMS) based on methods described previously [15]. Briefly, CE-FTMS analysis was carried out using an Agilent 7100 CE capillary electrophoresis system equipped with a Q Exactive Plus (Thermo Fisher Scientific Inc., Waltham, MA, USA), an Agilent 1260 isocratic HPLC pump, an Agilent G1603A CE-MS adapter kit, and an Agilent G1607A CE-ESI-MS sprayer kit (Agilent Technologies, Inc., Santa Clara, CA, USA). Systems were controlled by Agilent Mass Hunter workstation software LC/MS data acquisition for 6200 series TOF/6500 series Q-TOF version B.08.00 (Agilent Technologies) and Xcalibur (Thermo Fisher Scientific) and connected by a fused silica capillary (50 μm i.d. × 80 cm total length) with commercial electrophoresis buffers (H3301-1001 and I3302-1023 for cation and anion analyses, respectively, HMT) as the electrolytes. The spectrometer was scanned from *m*/*z* 60 to 900 in positive-ion mode and from *m*/*z* 70 to 1050 in negative-ion mode [15]. Peaks were extracted using Master Hands ver.2.19.0.2 automatic integration software (Keio University, Tsuruoka, Japan) in order to obtain peak information, including *m*/*z*, peak area, and migration time (MT) [16]. Areas of annotated peaks were then normalized to internal standards and sample volume in order to obtain relative levels of each metabolite. Detailed settings of the CE-FTMS were reported previously [15]. HCA and PCA were performed using MATLAB (ver.7.14.2) and R (ver.3.5.2).

### 2.5. Detailed Analysis of Carnitine and Acylcarnitines

0.5 mL of methanol and 2 μL of internal standard reagent (NeoSMAAT AC) were added to frozen tissue (5–20 mg), homogenized, and centrifuged (15,000 rpm for 15 min at 4 °C). Supernatant was transferred to another tube and diluted 10-fold with water. After centrifugation (12,000 rpm for 10 min at 4 °C), supernatant was analyzed by LC-MS/MS.

A Nexera™ X2 system coupled with an LCMS™-8060NX triple quadrupole mass spectrometer (Shimadzu Corporation, Kyoto, Japan) was used for LC–MS/MS with a Sun Shell PFP column (50 × 2.1 mm, 2.6 μm), and the column oven and autosampler were set to 40 °C and 4 °C, respectively. Mobile phase A was 0.1% formic acid and 5 mmol/L ammonium formate in water, and mobile phase B was methanol. The pump was operated at 0.3 mL/min with gradient elution. Gradient elution was performed as follows: 0% B (0–1 min), 70% B (1.5 min), 85% B (4.5 min), 100% B (5.5–10 min), 0% B (10.1–16 min). The injection volume of samples was set at 0.2 µL. For MS analyses, temperatures of the interface, desolvation line, and heat block were set at 400, 250, and 400 °C, respectively. Flow rates of nebulizer gas, heating gas, and drying gas were set at 3, 15, and 3 L/min, respectively. Data were acquired using MRM mode.

### 2.6. Statistical Analysis

Analysis of metabolites measured by CE-FTMS was performed using Welch’s *t*-test. Analysis of carnitine and acylcarnitines measured by LC-MS/MS and qPCR results were statistically analyzed using the unpaired student T-test using GraphPad Prism, version 7.00. *p* < 0.05 was considered statistically significant.

## 3. Results

### 3.1. Evaluation of Mice and Renal Fibrosis

At the time of tissue sampling, the average body weight in the TAC group was 45.9 ± 0.8 g, whereas that in the control group was 47.0 ± 0.5 g. There was no significant difference between the two groups (*p* = 0.27). Masson trichrome staining showed blue staining in tubular and interstitial portions of renal tissue in the TAC group, showing tubular and interstitial atrophy and fibrosis (Figure 2A–D).

### 3.2. Quantitative PCR

To confirm injury and fibrosis of renal tissue, quantitative PCR (qPCR) was performed for *Actin alpha 2 (Acta2)* and *Transforming growth factor beta1 (Tgfb1)* as biomarkers of tissue fibrosis. *Kidney injury molecule-1 (Kim-1)* was employed as a biomarker of tubular damage. Although there was no significant difference in Tgfb1 between the two groups (*p* = 0.26), Acta2 and Kim-1 were significantly higher in the TAC group than in the control group (TAC group/control group, Acta2; 1.48, *p* = 0.02, Kim-1; 6.48, *p* = 0.01) (Figure 2E–G).

### 3.3. Metabolite Detection Results by Metabolome Analysis

Metabolomic analysis using CE-FTMS is presented here. Principal component analysis (PCA) is shown in Figure 3A. The first principal component score (PC1), the highest proportion of variance, comprised 23.9% of all metabolites detected. The second principal component score (PC2), the second highest proportion of variance, included 21.1% of all metabolites detected (Figure 3A). A heatmap displaying hierarchical clustering analysis (HCA) of metabolites is shown in Figure 3B. PCA and the heat map indicated differences in metabolite profiles between the TAC and control groups. In this metabolomic analysis, 545 metabolite peaks were detected (anionic mode:cationic mode = 305:240). Of the metabolites detected, 65 differed significantly between the two groups.

Thirty-two metabolites were significantly higher and 33 were significantly lower in the TAC group (Table 1 and Table 2). Of metabolites that showed significant changes, carnitine and carnitine-related metabolites accounted for eighteen. Carnitine-related metabolites represented 27.7% of all metabolites with significant differences, which is the highest share ratio of any metabolites (Figure 4). Carnitine and acylcarnitines (acetylcarnitine, propionylcarnitine, lauroylcarnitine, malonylcarnitine, tiglylcarnitine, and octanoylcarnitine) were significantly lower in the TAC group than in the control group (Figure 5, Table 2). Aspartic acid, which is produced from acetylcarnitine, was also significantly lower in the TAC group. Arginine, which is produced from aspartic acid, was lower in the TAC group, as were homoarginine, guanidinoacetic acid, creatine, and ornithine. As aspartic acid is also involved in pyrimidine metabolism, UMP was also significantly lower in the TAC group (Figure 5, Table 2).

Metabolite groups other than carnitine and related metabolites that also showed significant changes were as follows: four metabolites related to histamine, three antioxidants, and two metabolites related to the glycolytic system (Figure 6A–C). These accounted for 6.1%, 4.6%, and 3.1% of total metabolites, respectively, all with significant differences (Figure 4). Histamine and its related metabolites, 1-imidazoleacetic acid, 1-methyl4-imidazoleacetic acid, and imidazole4-methanol, were all significantly higher in the TAC group (Figure 6A). Among glycolytic metabolites, phosphoenolpyruvate was significantly lower in the TAC group, whereas lactate was significantly higher (Figure 6B).

### 3.4. Detailed Analysis of Carnitine and Acylcarnitine in Renal Tissue

Detailed analysis of carnitine and short-, medium-, and long-chain acylcarnitines in metabolites in renal tissue was performed using LC-MS/MS. Carnitine and 10 acylcarnitines were significantly lower in the TAC group. Acylcarnitines that were significantly lower in the TAC group included acetylcarnitine, propionylcarnitine, butyrylcarnitine, isovalerylcarnitine, capronylcarnitine, octanoylcarnitine, decanoylcarnitine, myristoylcarnitine, palmitoylcarnitine, stearoylcarnitine (Figure 7). No acylcarnitine was significantly higher in the TAC group. 

## 4. Discussion

TAC has two adverse effects on the kidney. Acute kidney nephrotoxicity of TAC is a reversible dysfunction, but chronic kidney nephrotoxicity of TAC is irreversible and causes changes such as fibrosis of renal tubules and interstitial areas [3]. Since TAC-induced nephrotoxicity of renal allotransplantation is chronic kidney injury, it is important to understand the pathophysiology and mechanism using a chronic TAC nephrotoxicity model. Although there has been only one report of metabolite analyses in multiple organs of mice treated with TAC for 2 weeks, only two metabolites showed significant changes in kidney tissue [17]. In addition, that report did not identify any histological changes associated with kidney injury. In our study, exposure to TAC caused fibrotic changes and significant increases in *Acta2*, a biomarker of renal fibrosis, and *Kim-1*, a biomarker of renal injury. In addition, O-Sulfotyrosine, which is not elevated in acute kidney injury, but is elevated in the course of chronic kidney disease, was significantly elevated in the TAC group in this study [18]. As 2.6 adult mouse-days (8–12 weeks of age) are equivalent to one human year, 28 mouse-days are almost equal to 10 human years [19]. These results suggest that renal tissues closely reflect the pathophysiology of chronic kidney injury resulting from TAC nephrotoxicity. This is the first analysis of metabolic changes in the kidneys resulting from TAC-induced chronic nephrotoxicity.

Our metabolomic analysis identified 65 metabolites that were significantly different in renal tissue. Of these metabolites, the largest proportion (~30%) were carnitine and carnitine-related metabolites. Most carnitine-related metabolites were significantly lower in the TAC group. Detailed analysis using LC-MS/MS also showed that carnitine and several acyl-carnitines were also significantly lower in the TAC group, indicating that low levels of carnitine and acylcarnitines in renal tissue were reproducible. Carnitine is important for transporting long-chain fatty acids into mitochondria and for promoting β-oxidation to provide energy for cells and tissues.

Carnitine prevents mitochondrial dysfunction, making it an essential metabolite to maintain a proper biological environment for cells and tissues [20,21]. Furthermore, carnitine deficiency promotes fibrosis of renal tissue [22], and carnitine supplementation in rats with TAC-induced nephrotoxicity inhibited renal tissue fibrosis and prevented apoptosis [23]. Therefore, it is suggested that carnitine deficiency in renal tissue is crucial to understanding the pathogenesis of TAC nephrotoxicity and therapeutic intervention. Our finding of low levels of both carnitine and acylcarnitines suggests a pathological imbalance in the supply and demand of carnitine in renal tissues in the TAC group. There are three routes of carnitine supply to the body: (1) dietary, (2) biosynthetic, and (3) reabsorption from proximal renal tubules [24]. Since there was no difference in body weight at the time of kidney collection between the two groups, there were no obvious differences in food consumption or nutritional status. Interestingly, γ-butyrobetaine, the precursor of carnitine, did not differ between the two groups (Appendix A). Conversion of γ-butyrobetaine to carnitine requires hydrolysis by gamma-butyrobetaine hydroxylase 1 (BBOX-1) [25,26]. It is assumed that BBOX-1 function is impaired by TAC. In addition, since reabsorption of carnitine from urine into the cytoplasm is mediated by Organic Cation Carnitine Transporter 2 (OCTN2) in the proximal tubular epithelium [27], it is possible that inhibition of OCTN2 activity by TAC may cause carnitine depletion, tubular damage, and renal fibrosis.

Our results also showed a decrease in acetylcarnitine. Acetylcarnitine is involved in synthesis of aspartic acid [28], suggesting that the decrease in acetylcarnitine results in a decrease in aspartic acid. As a result, metabolites such as homoarginine, arginine, ornithine, creatine, and UMPs, all of which are involved in synthesis of aspartic acid, were presumably also significantly lower in the TAC group [29].

Several reports have comprehensively analyzed metabolites in patients with chronic kidney disease and in mouse models of chronic kidney disease, but none noted significantly lower levels of carnitine [10,30,31]. This suggests that low carnitine levels are specific to TAC nephrotoxicity. Therefore, carnitine and acyl-carnitines are potential biomarkers and targets for therapeutic intervention in TAC nephrotoxicity. Detailed investigation of the relationship between TAC and carnitine is required.

In addition to carnitine, significant changes were observed in glycolytic, antioxidant, and histamine-related metabolites. Although no significant changes were observed in metabolites of the citric acid cycle, lower phosphoenolpyruvate and higher lactate were seen in the TAC group (Figure 6C). In mitochondrial disease, mitochondrial dysfunction leads to anaerobic metabolism and increased lactate levels [32]. Therefore, elevated lactate detected in this study may indicate decreased mitochondrial function due to low carnitine levels. Indeed, there is a report that improving mitochondrial function in TAC nephrotoxicity reduces tissue damage [33].

The antioxidant metabolites, S-carboxymethylcysteine and N-acetylcysteine were lower in the TAC group (Figure 6B). These two metabolites are antioxidants that reduce tissue damage caused by reactive oxidative stress (ROS) [34,35], and their low levels may signify ROS-induced damage in the kidney. On the other hand, homocysteic acid was high in the TAC group, suggesting ROS overproduction [36]. The decreased antioxidant function in renal tissues suggests that a tissue deficiency of carnitine also affects mitochondrial metabolism and function.

Histamine and histamine-related metabolites were significantly elevated in the TAC group (Figure 6A). It has been reported that histamine-releasing mast cells increase in concert with decreased renal function [37]. In addition, histamine receptors exist in the kidneys, with H1 and H4 receptors mainly in renal tubules [38]. There are reports that activation of H4 receptors indicates a profibrotic response to renal damage and that H4 receptor agonists promote renal fibrosis [39,40]. Furthermore, histamine’s effect on H4 receptors enhances ROS production by phagocytic cells. Elevation of histamine and related metabolites in the present study may have contributed to H4 receptor-mediated tissue fibrosis and impaired antioxidant function in renal tissues in the TAC group. Further studies are warranted.

Since this is an in vivo experiment using mice, it is necessary to evaluate whether the renal pathology reported here reflects real-world clinical conditions. In fact, kidney transplantation recipients receive a much lower dose of tacrolimus than did mice in this study. On the other hand, mice in this study were healthy, without having undergone transplantation, which differs from the clinical situation. In addition, it must be determined whether carnitine and related metabolites are similarly produced in human samples. Although TAC nephrotoxicity is thought to be exacerbated by prolonged and sustained exposure to TAC, these results only show tissue changes after 4 weeks and do not track metabolic changes over time. Therefore, it is necessary to illuminate the pathogenesis of TAC nephrotoxicity by focusing on metabolic changes that occur with different periods of TAC exposure.

## 5. Conclusions

In this study, we used metabolite analysis to discover potential mechanisms of TAC nephrotoxicity. Carnitine deficiency in renal tissue is likely responsible for TAC nephrotoxicity.

## Figures and Tables

**Figure 1 biomedicines-12-00521-f001:**
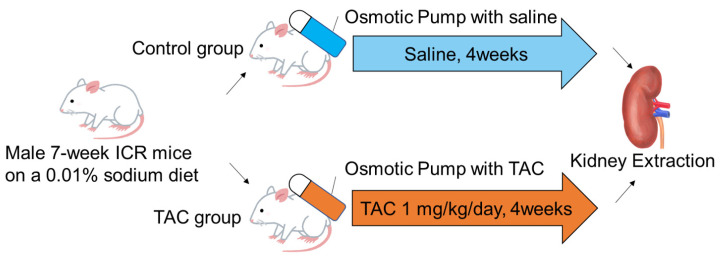
TAC administration protocol for mice. Seven-week ICR mice were divided into control and TAC groups. Both groups were fed a low sodium diet (0.01%) for 7 days before drug administration. The osmotic pump (ALZET^®^ osmotic pump 2004) of the TAC group was filled with TAC equivalent to 1 mg/kg/day for 40 h prior to drug administration, and the pump of the control group was filled with saline. After incubation, osmotic pumps were implanted subcutaneously in the back of each mouse in both groups, and continuous subcutaneous administration was performed for 28 days. After administration, kidney samples were collected under general anesthesia. ICR: Institute of Cancer TAC: Tacrolimus.

**Figure 2 biomedicines-12-00521-f002:**
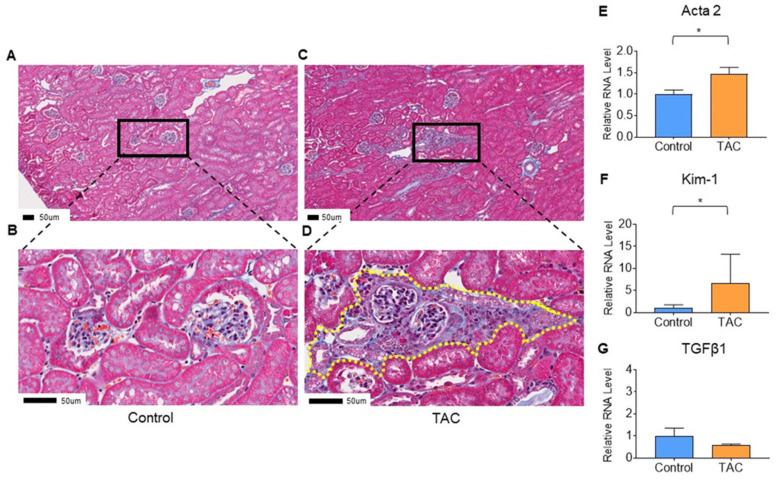
Evaluation of renal tissue damage. (**A**–**D**) show the results of Masson trichrome staining of kidney samples. These are representative pathological findings for each group. (**A**,**B**) are pathological findings of the Control group, and (**B**) is a magnified image of (**A**). (**C**,**D**) are pathological findings of the TAC group, and (**D**) is an enlarged image of (**C**). Yellow dotted line in (**D**) shows the tubular and intestinal fibrosis. All scale bars are 50 μm (n = 5/group). Quantitative PCR (qPCR) results are shown in (**E**–**G**). (**E**) *Actin alpha 2 (Acta2)*, (**F**) *Kidney injury molecule-1 (Kim-1)*, and (**G**) *Transforming growth factor beta1 (Tgfb1)*. Statistical analysis was performed using an unpaired *t*-test. (* *p* < 0.05, n = 10/group).

**Figure 3 biomedicines-12-00521-f003:**
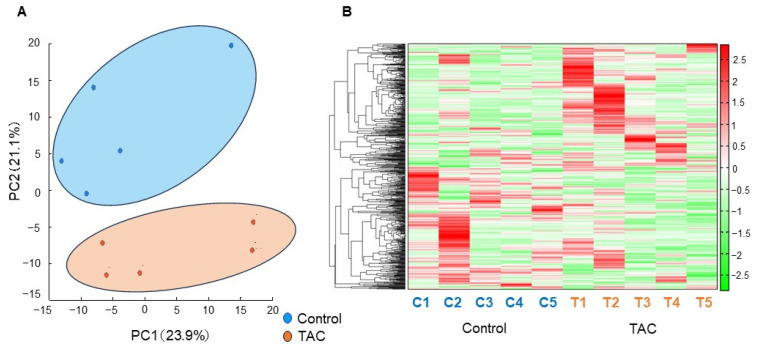
Principal Component Analysis and Heatmap of Hierarchical Clustering Analysis. 545 metabolites in renal tissue were examined in our metabolomic analysis. A clear difference was found between the TAC and control groups with principal component analysis (PCA) and a heatmap displaying hierarchical clustering analysis (HCA). (**A**) Results of PCA. The horizontal axis indicates the first principal component score (PC1), and the vertical axis shows the second principal component score (PC2). The control group (blue), and the TAC group (orange) differed significantly in metabolite profiles. (n = 5/group). (**B**) Heatmap displaying HCA of metabolites with a tree of sequences. The bar on the right side of the heatmap shows the relationship between the measured values and the color of each sample. Red indicates values higher than average, and green denotes values lower than average. The vertical axis shows metabolites. The horizontal axis identifies samples C1 to C5 in the Control group, and T1 to T5 indicates individuals in the TAC group (n = 5/group).

**Figure 4 biomedicines-12-00521-f004:**
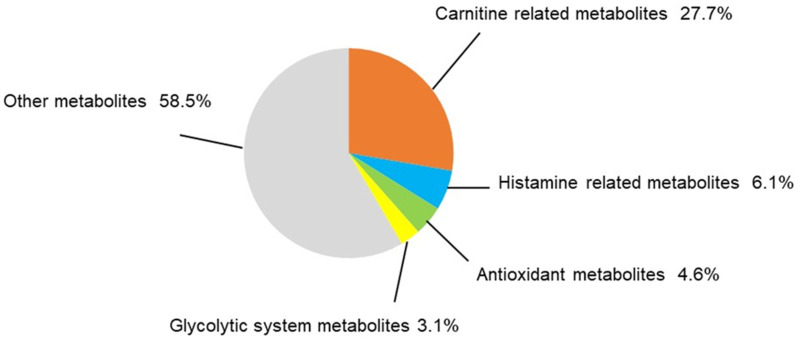
Categorization of significantly different metabolites. In this metabolomic analysis, 65 metabolites differed significantly between the two groups. The share ratio of each metabolite group is represented here. Statistical analysis was performed with Welch’s *t*-test. *p* < 0.05 was considered statistically significant. (n = 5/group).

**Figure 5 biomedicines-12-00521-f005:**
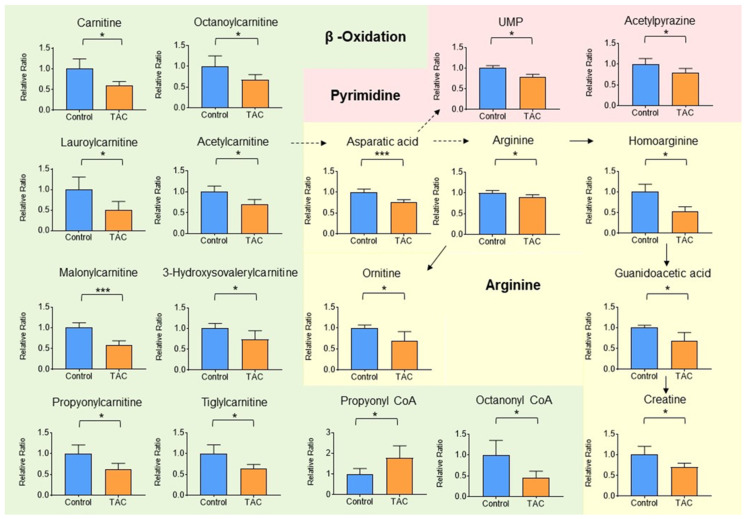
Carnitine-related metabolites measured by metabolome analysis. Orange columns represent the TAC group, and blue columns represent the Control group. The vertical axis is the relative ratio. The ratio was calculated as follows. The relative ratio = normalized ratio of Tac group/normalized ratio of control group. Solid lines indicate relationships between metabolites and their precursors. Dotted lines indicate relationships involved in metabolite production. Among metabolites related to carnitine, metabolite groups involved in β-oxidation and mitochondria are shown in green. Those related to pyrimidine metabolism in pink, and those related to aspartic acid and arginine are shown in yellow. (* *p* < 0.05, *** *p* < 0.001) (n = 5/group).

**Figure 6 biomedicines-12-00521-f006:**
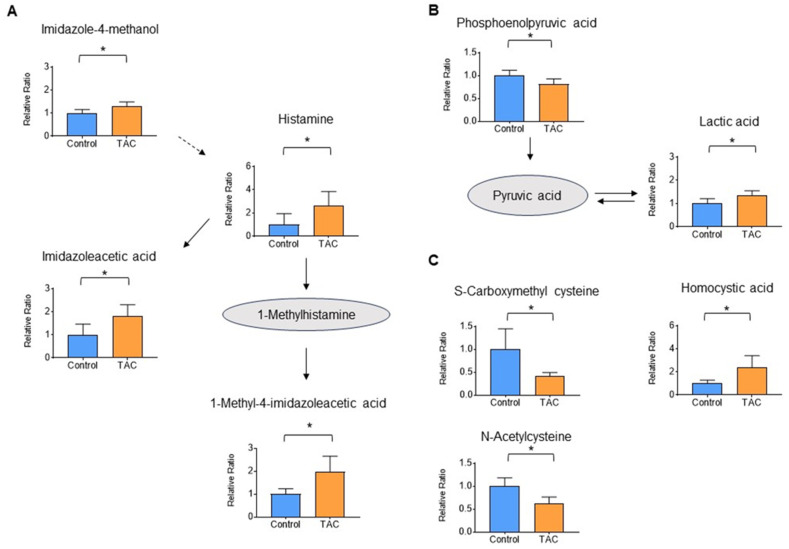
Histamine, glycolytic, and antioxidant metabolites measured by metabolome analysis. Orange columns represent the TAC group, and blue columns represent the Control group. The vertical axis is the relative ratio. Solid lines indicate the relationship between metabolites produced and precursors. Dotted lines indicate relationships for metabolite production. (**A**) Metabolites related to the histamine pathway (**B**) Metabolites related to the glycolytic pathway. (**C**) Metabolites related to antioxidants (* *p* < 0.05) (n = 5/group).

**Figure 7 biomedicines-12-00521-f007:**
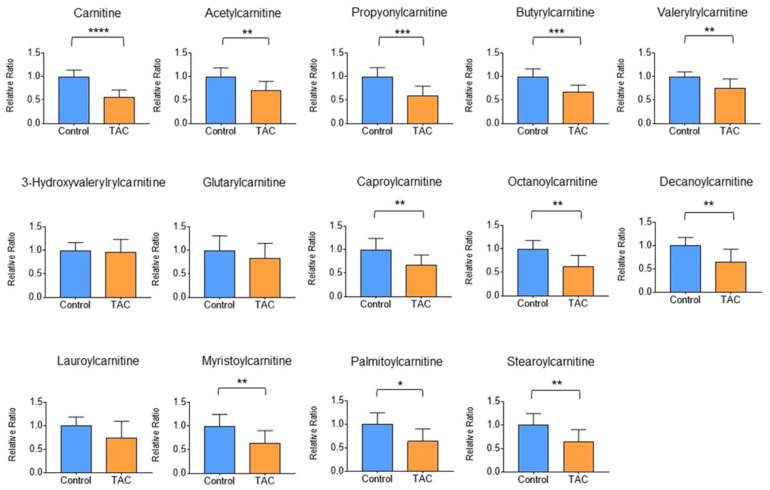
Detailed analysis of carnitine-related metabolites. Comparisons of carnitine and acylcarnitines measured by LC-MS/MS between the two groups are shown here. Orange columns indicate the TAC group and blue columns show the control group. The mean of the control group is taken as 1 and the mean of the TAC group is relative to the control group. The vertical axis is the relative ratio. (* *p* < 0.05, ** *p* < 0.01, *** *p* < 0.001, **** *p* < 0.0001) (n = 9–10/group).

**Table 1 biomedicines-12-00521-t001:** Tissue metabolites significantly more abundant in the TAC group.

Tissue Metabolites	Ratio	*p*-Value
Carboxymethyllysine	1.3	0.040
myo-Inositol 2-phosphate	1.3	0.010
Imidazole-4-methanol	1.3	0.043
3-Guanidinopropionic acid	1.3	0.034
Lactic acid	1.3	0.039
H-Asp(Gly-OH)-OH	1.4	0.004
Sedoheptulose 7-phosphate	1.4	0.042
Thymine	1.4	0.030
N-Acetylthreonine	1.5	0.026
N-Acetyltaurine	1.5	0.030
γ-Carboxyglutamic acid	1.5	0.001
Sulfotyrosine	1.6	0.003
2′-Deoxycytidine	1.6	0.006
Orotidine	1.7	0.030
N-Ethylmaleimide_+H O_2_	1.7	0.046
Propionyl CoA_divalent	1.8	0.039
Imidazolelactic acid	1.8	0.032
Nicotinamide riboside	1.8	0.039
N-Acetylalanine	1.8	0.023
Acetylcholine	1.8	0.032
Glucaric acid	1.9	0.002
1-Methyl-4-imidazoleacetic acid	2.0	0.030
1-Methylnicotinamide	2.0	0.036
Ascorbate 2-sulfate	2.3	0.034
Dihydroxyacetone phosphate	2.3	0.027
Homocysteic acid	2.4	0.036
Histamine	2.6	0.050
1-Methyladenosine	2.8	0.048
2-Aminoisobutyric acid	2.9	0.020
3-Methylcytidine	2.9	0.043
cCMP,2′,3′-cCMP	3.0	0.017
threo-β-Methylaspartic acid	21	0.027

Ratio = normalized ratio of TAC group/normalized ratio of control group.

**Table 2 biomedicines-12-00521-t002:** Tissue metabolites significantly lower in the TAC group.

Tissue Metabolites	Ratio	*p*-Value
S-Carboxymethylcysteine	0.4	0.041
Octanoyl CoA_divalent	0.5	0.023
Ethyl glucuronide	0.5	0.020
Lauroylcarnitine	0.5	0.018
Homoarginine	0.5	0.002
Malonylcarnitine	0.6	0.0
Capryloylglycine	0.6	0.037
Carnitine	0.6	0.017
Propionylcarnitine	0.6	0.014
N-Acetylcysteine	0.6	0.007
Tiglylcarnitine	0.6	0.015
Ala-Ala	0.6	0.026
3-Hydroxykynurenine	0.7	0.033
Gamma-Glu-Asp	0.7	0.0007
Octanoylcarnitine	0.7	0.043
Homoserine	0.7	0.005
Ornithine	0.7	0.034
Guanidoacetic acid	0.7	0.023
O-Acetylcarnitine	0.7	0.006
Creatine	0.7	0.027
N -Acetyllysine	0.7	0.009
3-Hydroxyisovalerylcarnitine	0.7	0.050
Asparatic acid	0.8	0.0007
Citramalic acid	0.8	0.047
AMP	0.8	0.020
UMP	0.8	0.001
GDP-mannose, GDP-glucose	0.8	0.003
FAD_divalent	0.8	0.037
Pyridoxamine 5′-phosphate	0.8	0.010
Acetylpyrazine	0.8	0.028
Phosphoenolpyruvic acid	0.8	0.037
Arginine	0.9	0.033
UDP-N-acetylglucosamine	0.9	0.042

Ratio = normalized ratio of TAC group/normalized ratio of control group.

## Data Availability

The datasets generated and/or analyzed during the present study are available from the corresponding author on request.

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
