# Peer review of "Metabolomic Profiling of Mice with Tacrolimus-Induced Nephrotoxicity: Carnitine Deficiency in Renal Tissue"

_biomedicines, 2024, doi:10.3390/biomedicines12030521_

Round 1
Reviewer 1 Report
Comments and Suggestions for Authors
This manuscript investigates Tacrolimus (TAC) induced nephrotoxicity using a metabolomics approach. 545 metabolites were examined in kidney tissues. The toxicity model used was the low sodium diet model in mice, which have been treated with TAC at 1 mg/kg/day for the duration of 4 weeks. The manuscript is well written and has scientific merit.
There are some major concerns that should be addressed:
Major concerns:
· Sham group is missing. In the context of metabolomics analysis it would be essential to investigate if a low salt/ low sodium diet has an effect on carnitine metabolism. Carnitine and its metabolites are important osmolytes with metabolic functions. A low sodium model might impact and precondition the osmolyte balance. Thus, it is important to understand what effects the low sodium model in comparison to normal diet animals (sham) might have for the proper interpretation of the effects of TAC.
· Statistics and group size. There is no justification for the relatively low number of animals per study group (n=5). Was a Power Analysis performed and what were the parameters? Was a Bonferroni correction performed?
I have the following minor comments/suggestions:
· Materials and Methods: The information regarding the capillary electrophoresis settings are not listed. This should be available either in the manuscript or supplementary information section.
· Materials and Methods: It is not clear to the reader what internal standard compounds were used. There is a product number “H3304-1002” listed, but further information regarding the internal standards used is not available. This is essential information, since all data were normalized to the internal standards. This information should be available either in the manuscript or supplementary information section.
· Discussion: There is a recent publication by Aouad et al. (2023) that showed that carnitine was significantly increased in cells treated with TAC (n=9 per study group). These findings should be discussed and interpreted as well. Is there a difference between the acute and chronic toxicity?
Reviewer 2 Report
Comments and Suggestions for Authors
Nishida et al. conducted a comprehensive analysis of kidneys exposed to Tacrolimus, employing histopathological examination, PCR, and metabolome analysis. Their findings revealed distinct tubular and interstitial atrophy and fibrosis, accompanied by elevated levels of Actin alpha 2 (Acta2) and Kidney injury molecule-1 (Kim-1) in mice treated with Tacrolimus. Notably, the Tacrolimus-treated group exhibited a significant increase in 32 metabolites and a decrease in 33 metabolites.
I propose the following modifications and additions:
· Could you provide information on the number of animals per group used for histopathological (HP) analysis?
· Existing literature suggests that transplant patients typically take substantially lower doses of tacrolimus orally (and low bioavailability, while during parenteral treatment bioavailability is high), around 10-fold less. Could you provide a reference supporting the assertion that 1 mg/kg/day is not a toxic dose? The study by Xie D, Guo J, Dang R, Li Y, Si Q, Han W, Wang S, Wei N, Meng J, Wu L (BMC Pharmacol Toxicol. 2022 Nov 28;23(1):87. doi: 10.1186/s40360-022-00626-x.) utilized doses that were circa threefold lower. I can’t find that any study which used such a high concentrations of tacrolimus.
· It is crucial to note that this study examined the effects of Tacrolimus on healthy kidneys that were not transplanted. Unlike the healthy mice in this study, individuals in the terminal phase of chronic kidney disease (CKD) are often debilitated, with multiple affected organs, including compromised heart and liver functions. Please acknowledge this in the limitations section to clarify that the effects of Tacrolimus were assessed in healthy mice.
· Consider adding the number of mice used per group to the abstract.
· Commendable figures and an appropriately structured paper. The discussion is well-written.
· The examination that authors conducted are inaccessible to physicians and therefore it is valuable that we are aware of pathophysiological background of TAC-induced nephrotoxicity but we are missing some examination assessable to physicians (e.g. serum levels of IL-2) that might correlate with metabolome or HP findings so, doctors may have insights regarding potential extent of fibrosis or metabolome changes in individuals who transplanted, when they measure e.g. IL-2 levels etc.
· May authors calculate how long would 28 mice days’ transposases in human year. It would be nice to known after which period of time, adverse changes in kidney would certainly appear.
Round 2
Reviewer 2 Report
Comments and Suggestions for Authors
I thank the authors for careful revision and attention to details. Well done. I would like to suggest authors insert comment number 7 in the text of the manuscript that 28 mice days are equivalent to 10 human years.